# Experimental vs. Theoretical Viscosity Determination of Aluminosilicate Glasses

**DOI:** 10.3390/ma16175789

**Published:** 2023-08-24

**Authors:** Anna Zawada, Malgorzata Lubas, Adrian Nowak

**Affiliations:** Department of Materials Engineering, Czestochowa University of Technology, Armii Krajowej 19, 42-200 Czestochowa, Poland; malgorzata.lubas@pcz.pl (M.L.); adrian.nowak@pcz.pl (A.N.)

**Keywords:** viscosity, aluminosilicate glasses, melting process, natural raw materials, glass cullet, theoretical calculation methods

## Abstract

The paper presents the results of studies on the viscosity of the glass mass in various temperature ranges, determining the basic technological parameter, very important from the point of view of melting and forming. For this purpose, six sets based on natural raw materials such as basalt, dolomite, and amphibolite, modified with different amounts of float glass cullet, were melted. The melting process was carried out in an electric furnace at the temperature of 1450 °C for 2 h. Using the dilatometric method, high-temperature microscopy and theoretical calculation methods, the viscosity of the produced glasses was determined in various temperature ranges. Comparative analyses of the employed methods were carried out. The significance of the applied calculation methods for aluminosilicate glasses depending on the basic chemical composition of the glasses was presented. The relationship between the manner of incorporating amphoteric ions Al^3+^, Fe^3+^ and Mg^2+^ into the glass structure and the change in viscosity in the temperature range corresponding to the working point range at 10^4^ [dPa·s] viscosity and the relaxation range—*T_g_* temperature at 10^13^ [dPa·s] viscosity was justified. It was justified that in order to plot the viscosity curve with the correct slope in the forming range for aluminosilicate glasses, it is appropriate to use the two-point method based on the fixed viscosity points of 10^4^ [dPa·s] and 10^13^ [dPa·s].

## 1. Introduction

Knowledge of the basic technological parameters in the production of each type of glass products has a significant impact on the entire organization of the process of melting, clarifying, forming and annealing the glass mass. One of the most important parameters having a significant impact on the quality of the manufactured products is the viscosity of the glass mass. This property determines the formation of an amorphous glass structure, which in relation to the glassy state is of great theoretical and practical importance [1]. From a technological point of view, due to the high energy consumption of the entire process of melting the batch of raw materials and then forming the glass mass, a thorough analysis of the dependence of the glass mass viscosity on the temperature becomes justified [2,3]. Owing to the type of melted glass, the viscosity of the glass mass can change in different temperature ranges. The basic chemical composition of the glass mass has a significant impact on the course of viscosity change, i.e., on the so-called viscosity curve. The chemical composition of the glasses is also the basis for choosing their forming method, e.g., float glass (soda–lime–silicate glass), mineral fibers (aluminosilicate glass), laboratory vessels (borosilicate glass) [4,5,6,7,8,9]. The high viscosity of the glass means that the process of melting and clarifying the mass must be carried out in an increased temperature range, which in turn generates higher production costs. For economic reasons, in order for the entire technological process to be conducted in optimal conditions, it is necessary, among others, to monitor the stability of the correct temperature at each stage of glass production. Due to the significant importance of the viscosity of the glass mass in the entire production cycle, many theoretical studies have been performed to analyze the impact of various factors on changes in the viscosity of glasses in specific temperature ranges [10]. Among others, models and mathematical relationships have been developed to describe viscosity as a function of temperature, e.g., the Vogel–Fulcher–Tammann (VFT) model [11,12], the Tuszynski equation [13], the M.W. Ochotin model and others [14,15]. One of the best-known and most commonly used models is the Vogel–Fulcher–Tammann (VFT) model. It is utilized to determine the viscosity of most industrial glasses in a wide temperature range, e.g., from ambient temperature to melting and homogenization temperatures [16,17,18]. The use of this model is very convenient because it is based on the transition temperature, *T_g_*, characteristic for each type of glass, with its constant viscosity *η* = 10^13^ [dPa·s]. Taking into account the value of temperature *T_g_*, it is possible to determine the course of the viscosity curve in any temperature range. Unfortunately, the use of this model in the case of borosilicate glasses, in which the content of alkali oxides exceeds 15%, is not possible because with the increase in their content, the properties of these glasses change in a non-linear way (borate anomaly). Similarly, in the case of aluminosilicate glasses, the Vogel–Fulcher–Tammann model will not be applicable. For this type of glass, depending on the content of various oxides, including mainly amphoteric oxides (Al_2_O_3_, MgO or Fe_2_O_3_), the properties do not change linearly either, but rather exponentially, which means that for glasses which are extremely different in terms of chemical composition, it is possible to obtain similar properties, e.g., transition temperature *T_g_* [19,20,21].

Therefore, it is necessary to intensify research and analysis for this type of glass in terms of optimizing the process of melting the glass mass and improving the quality of the formed products. As part of this work, the viscosity of aluminosilicate glasses from the SiO_2_–Al_2_O_3_–Fe_2_O_3_–CaO–MgO system, melted from mineral resources such as amphibolite and basalt with the addition of dolomite and float glass cullet, was analyzed.

Experimental methods were employed to determine the viscosity curves, i.e., dilatometric analysis, by means of which transition temperature *T_g_* was determined at a glass viscosity of 10^13^ [dPa·s], and the dilatometric softening point (*T_d_*) at a viscosity of 10^12^ [dPa·s]; in addition, the high temperature microscopy method was employed, by means of which the temperature of the hemisphere was determined at a glass viscosity of approx. 10^4^ [dPa·s]. Ochotin and Tuszynski and Dietzel–Brückner calculation methods were also used [22,23,24].

## 2. Materials and Methods

The research material consisted of aluminosilicate glasses obtained from the main mineral raw materials, i.e., amphibolite (Pilawa Gorna mine) and basalt (Wilkow mine) as well as raw materials modifying glass sets, i.e., dolomite (Redziny—Lower Silesia) and float cullet. Six sets of raw materials were prepared for melting, the chemical composition and the percentage composition of which is shown in Table 1.

The weighed sets of raw materials were homogenized and melted in a platinum crucible in an electric furnace (Czylok-Polska) at a temperature of 1450 °C for 2 h, and then poured onto a steel plate. Analysis of the basic chemical composition was carried out for the obtained glasses by means of XRF spectroscopy. The study was performed using a WDXRF Axios mAX spectrometer with a 4 kW Rh lamp from PANalytical. The dilatometric test was performed utilizing a Sadamel DA-3 linear dilatometer, in an air atmosphere, at a furnace heating rate of 10 K/min. This test was conducted on rod-shaped glass samples with a diameter of about 3 mm and length of 10 mm. Hot-stage microscopy analysis was carried out on a Hesse Instruments microscope in air atmosphere, with a furnace heating rate of 10 K/min. Crushed glass (fraction below 63 μm) was used for the measurement, from which cylinders with a diameter and height of 3 mm were formed, then placed on a corundum plate and the measurement was carried out.

## 3. Results and Analysis

### 3.1. Determination of Chemical Composition

The basic chemical composition of the obtained glasses was determined using XRF spectroscopy, and the results of the analysis are presented in Table 2.

On the basis of the analysis of the chemical compositions of all the obtained glasses, the ranges of percentages of individual oxides were determined, which are presented in Table 3.

### 3.2. Determination of Temperatures for Constant Viscosity Values by Experimental Methods

#### 3.2.1. Dilatometric Test

The dilatometric method was employed to determine transition temperature *T_g_*, at which the glass viscosity was 10^13^ [dPa·s]. The measurement was carried out for all the glasses, and then on the obtained dilatometric curves (Figure 1), the *T_g_* temperature and dilatometric softening point *T_d_* at a viscosity of about 10^12^ [dPa·s] were determined graphically [25,26]. The test results are presented in Table 4.

By analyzing the determined values of the transition temperatures and dilatometric softening of the examined glasses, it can be seen that the glass obtained from Set 1 had the highest temperatures, while the glass melted from Set 2 had the lowest temperatures (Table 1). Similar temperatures of *T_g_* and *T_d_* to the temperatures of the melted glass from Set 1 were also obtained for the glasses from Sets 4 and 5. Taking into account the individual chemical compositions of all the glasses, it is difficult to unambiguously assess the effect of the additives, i.e., dolomite, glass cullet, on the viscosity of the obtained glasses. Interpretation of the determined temperatures is difficult because of the fact that the basic chemical composition of the glasses contains oxides of amphoteric metal ions, i.e., Al, Fe and Mg, which can obtain a coordination number of both 6 and 4 in the glass structure [20,27,28,29,30]. This means that they can act as modifying ions (coordination number—CN 6) or build into the network as network-forming ions (CN 4). In the case of glasses containing a high proportion of metal ions in their chemical composition, which may occur in variable coordination, each increase in the additional amount of free oxygen in the form of O^2−^ ions in the glass mass, e.g., by adding alkaline cullet (the introduction of alkali oxides), will cause an increase in the number of amphoteric metal ions in the lower coordination. The increase in metal ions in coordination number 4 leads to the incorporation of these ions into the glass network, and thus its strengthening, which in turn raises the melt viscosity [31]. Such examples are Glasses 3 and 6 and Glasses 4 and 5, whose *T_g_* temperature reached similar values, although the composition of Glasses 5 and 6 contained the highest content of alkali oxides (a high share of glass cullet in the set, respectively: 20% and 30%).

By analyzing the determined values of the *T_g_* and *T_d_* temperatures, it can be concluded that the modification of amphibolite glass (Set 1 *T_g_* = 661 °C) with the addition of 10% dolomite and 20% glass cullet (Set 2) causes a significant decrease in the viscosity of the glass mass (*T_g_* = 600 °C), and thus easier melting of the raw set material [32]. With the modification of amphibolite glass with a smaller amount of dolomite (5%) and glass cullet in the amount of 10%, Glass 3 (*T_g_* = 639 °C) did not have such a significant effect on reducing the viscosity of the glass as was in the case of Glass 2. The viscosity of the glass melted from Set 3 (85 wt% amphibolite, 5 wt% dolomite, 10 wt% cullet) can be compared with the viscosity of Glass 6 (*T_g_* = 641 °C) obtained from the set: 20 wt% amphibolite, 50 wt% basalt, 30 wt% cullet. Despite significant differences in the basic chemical composition of the glasses, they had a similar transition temperature. A similar relationship can be observed in Glasses 4 (*T_g_* = 657 °C) and 5 (*T_g_* = 658 °C). The increase in the share of alkali oxides in Glass 5 did not cause a significant decrease in its viscosity. This fact is confirmed by the statement that the presence of alkali metal oxides (i.e., Na_2_O and K_2_O) has a significant impact on the viscosity of aluminosilicate glasses containing a large proportion of metal ions that can assume coordination 4 and 6 (e.g., Al^3+^, Fe^3+^, Mg^2+^) [30]. The presence of these modifiers in the glass mass increases the number of free oxygen ions, enabling the amphoteric ions to achieve tetrahedral coordination and incorporate them into the glass network, which results in an increase in the viscosity of the glass mass [20,21]. The same trend can be observed in determining the temperature for dilatometric softening point *T_d_* (*η* = 10^12^ [dPa·s]).

#### 3.2.2. High Temperature Microscopy

On the basis of the high-temperature analysis, characteristic temperatures were determined, accompanied by a change in the shape of the sample depending on the viscosity of the glass. This study allowed the temperatures to be determined at glass viscosity *η* defined in [dPa·s]: deformation of *logη* = 6.3, sphere of *logη* = 5.4, hemisphere of *logη* = 4.1 and flow of *logη* = 3.4 [33,34]. The temperatures for the characteristic (constant) viscosity values of the researched glasses are presented in Table 5 and Figure 2.

The values of the characteristic glass temperatures obtained in the high temperature tests reveal a different temperature dependence than in the case of the dilatometric tests. At higher temperatures, with viscosity in the range of 10^3^–10^7^ [dPa·s], called the working range, the highest temperature values were obtained for the glass melted from Set 1 (100% amphibolite: 875–1140 °C), and the lowest from Set 6 (20% amphibolite, 50% dolomite, 30% cullet: 784–1010 °C). This means that in the range of high temperatures, diffusion grows and the mobility of ions in the glass mass increases, which results in a decline in its viscosity. In this case, the modifying effect of basalt and cullet on the viscosity of the glass mass is visible, consisting in lowering the temperatures in the molding range [35,36].

### 3.3. Calculation Methods

All the methods of calculating the viscosity of glasses are based on their basic chemical composition. The influence of individual oxides on the viscosity of the glass mass is conditioned by the way individual metal ions are incorporated into the amorphous structure of the glass. The high proportion of ions forming the glass network will significantly increase the viscosity of the glass, while the growth in the content of modifying ions will have a diverse effect on it in specific temperature ranges. In order to determine the viscosity of the studied glasses by computational methods, the following methods were selected: Vogel–Fulcher–Tammann, M.W. Ochotin, Tuszynski and Dietzel-Brückner. These methods were used to determine the transition temperature, *T_g_*, which is a fixed point on the viscosity curve at the value of *η* = 10^13^ [dPa·s] and the working point as the temperature value at the viscosity of *η* = 10^4^ [dPa·s]. The working point describes the temperature at which the glass is soft enough to be processed in glassworks by the most common methods such as blowing, pressing or drawing [1].

#### 3.3.1. Viscosity—Vogel–Fulcher–Tammann Method

The viscosity of glasses can be determined mathematically using Equation (1) given by Vogel–Fulcher–Tammann, called the VFT equation in logarithmic form [37,38]:(1)$$log\;\eta =A+\frac{B}{T-T_{0}}$$
where *A*, *B*, *T*_0_ are constants depending on the chemical composition of the glass. Based on Equation (1), the temperatures corresponding to the given viscosity value were calculated (Table 6).

Based on the obtained results, it can be clearly stated that this method cannot be employed to determine the viscosity of aluminosilicate glasses. The calculated temperatures drastically differ from the actual ones determined by experimental methods. The temperature values obtained in the calculation method clearly indicate the need to take into account the share of amphoteric oxides (e.g., Al_2_O_3_, Fe_2_O_3_, MgO) in the theoretical calculations, together with their ability to perform both bond-forming and modifying functions. The high-temperature values obtained in the calculations also result from the fact that the share of iron oxide was not included in the VFT equation. The investigated glasses from the SiO_2_–Al_2_O_3_–Fe_2_O_3_–CaO–MgO system were characterized by a high content of Fe_2_O_3_ oxide (from 7.31% by weight—Set 6 to 10.3% by weight—Set 1), which excludes the use of the Vogel–Fulcher–Tammann method in determining the viscosity curves of these glasses.

#### 3.3.2. Viscosity—M.W. Ochotin Method

The method according to Ochotin is used for glasses in which, apart from the main glass-forming oxide SiO_2_, there are also amphoteric oxides, i.e., Al_2_O_3_ and MgO, able to play a bond-forming role. This type of glass is mainly used in the production of fibers. This method makes it possible to calculate temperatures for characteristic viscosity values, important from the point of view of technological properties, with fairly high accuracy. These temperatures can be determined using Equation (2) [14]:*T* = *Ax* + *By* + *Cz* + *D*(2)
where

*T*—temperature [°C]*x*—percentage content of Na_2_O*y*—percentage content of the sum of CaO and MgO*z*—percentage content of Al_2_O_3_*A*, *B*, *C*, *D*—coefficients presented in Table 7

**Table 7 materials-16-05789-t007:** Coefficients for determining temperature [°C] at specified viscosities of 10^13^ and 10^4^ [dPa·s] according to M.W. Ochotin method.

Viscosity [dPa·s]	Coefficients			
	*A*	*B*	*C*	*D*
10^13^	−7.32	3.49	5.37	603.4
10^4^	−17.49	−9.95	5.9	1381.4

By means of Equation (2), the temperatures for the examined glasses were calculated at the specified viscosity values of 10^13^ and 10^4^ [dPa·s]. The results are presented in Table 8.

**Table 8 materials-16-05789-t008:** Calculated temperature [°C] at specified viscosities of 10^13^ and 10^4^ [dPa·s] according to M.W. Ochotin method.

Viscosity [dPa·s]	Glass					
	1	2	3	4	5	6
	Temperature [°C]					
10^13^	677	683	686	689	669	650
10^4^	1262	1134	1198	1175	1147	1118

Based on the obtained results, it can be seen that the temperature values at the viscosity of 10^13^ [dPa·s] are close to those obtained experimentally. Therefore, it is concluded that this method can be utilized to calculate the transition temperature *T_g_* for the researched aluminosilicate glasses.

#### 3.3.3. Viscosity—Tuszynski Method

In the Tuszynski method, to calculate the temperatures at the viscosities of 10^13^ and 10^4^ [dPa·s], the chemical compositions of the glasses listed in Table 2 were used. The calculations were carried out using Polynomial (3) [13].
*T* = *A*_1_*x*_1_ + *A*_2_*x*_2_ + *A*_3_*x*_3_ + *A*_4_*x*_4_ + *A*_5_*x*_5_ + …+ *A_n_x_n_*(3)
where: *A*_1_, *A*_2_, …, *A_n_*—calculation factors (Table 9), *x*_1_, *x*_2_, …, *x_n_*—mass percentage of oxides present in the glass.

The obtained results are summarized in Table 10.

Also in the case of this method, the *T_g_* temperature values obtained for the studied glasses are comparable to the experimental data, but with a tendency to underestimate them.

#### 3.3.4. Viscosity Calculated from Polynomial for Aluminosilicate Glasses

Using the calculation method developed by the authors of this work for glasses from the SiO_2_–Al_2_O_3_–Fe_2_O_3_–CaO–MgO system [19,39], based on the basic chemical composition of the investigated glasses, the temperatures at specific viscosities were calculated at *η* = 10^13^ and *η* = 10^4^ [dPa·s]. In the employed calculation method, the percentage content of all the oxides having a significant impact on the viscosity of the glasses was taken into account. The proposed algorithm works well for glasses whose basic oxide composition falls within the ranges presented in Table 11.

With the calculation method used, the obtained results will be close to the experimental values only when the percentage compositions of the tested glasses are close to the ranges in Table 11. In Table 3 shows the percentage ranges of oxide contents the tested glasses.

Observing the dependence of the glass properties on the basic chemical composition, a second-order computational prediction method in the form of Equation (4) was assumed.
*y* = *b*_1_*x*_1_ + *b*_2_*x*_2_ + *b*_3_*x*_3_ + *b*_4_*x*_4_ + *b*_5_*x*_5_ + *b*_6_*x*_6_ + *b*_7_*x*_1_*x*_2_ + *b*_8_*x*_1_*x*_3_ + … + *b*_21_*x*_5_*x*_6_(4)
where *y* is the property, *b*_1…_*b*_21_ are parameters, *x*_1_…*x*_6_ are the composition in weight % of the glasses.

Table 12 presents the temperatures calculated according to the polynomial for aluminosilicate glasses at the viscosities of *η* = 10^13^ and *η* = 10^4.0^ [dPa·s].

By comparing the *T_g_* temperature values obtained on the basis of calculations from all the selected calculation methods with the temperature values obtained from the dilatometric tests, it can be concluded that the calculation results that are the closest to the experimental ones were obtained utilizing the method developed by the authors of this article and by the Tuszynski method (Figure 3). Both of these methods take into account a high proportion of CaO oxide and amphoteric oxides, which can strengthen or modify the glass structure in various temperature ranges, both high (forming range) and low (relaxation range). Based on the analysis, it can be concluded that when using analytical methods, the appropriate calculation model should always be selected for the type of glass for which the calculations are carried out. In the case of aluminosilicate glasses, the influence of the chemical composition on the properties, both technological and functional, is quite different than in the case of soda–lime–silicate glasses. The temperatures calculated by the Vogel–Fulcher–Tammann method, *T_g_* and working point (Table 6), completely eliminate this type of calculation for applications for aluminosilicate glasses.

### 3.4. Viscosity Curve Plotting Based on Specific Temperatures at Viscosities of 10^13^ and 10^4^ [dPa·s]

The relationship between viscosity and temperature is very important in the production of glass. Hence, the basic feature of glass is a constant and smooth change in viscosity over the entire temperature range. In the range where the viscosity of the glass is 10^13^ [dPa·s], temperature *T_g_* is taken as the annealing temperature. This is crucial for stress relaxation after hot forming, indicating the upper temperature limit of the so-called the annealing range in which internal stresses are released within minutes. At higher temperatures, the characteristic temperature at the viscosity of 10^4^ [dPa·s] is the working point temperature. Depending on the size of the temperature range from 10^3^ to 10^7^ [dPa·s], a distinction is made between “long” glasses (large intervals between the softening point and the working point temperature) and “short” glasses (small intervals between the softening point and the working point temperature).

Knowing the values of the main temperatures, *T_g_* and the working point, it is possible to plot the viscosity curves over a wide temperature range [24].

#### 3.4.1. One-Point Method

In the one-point method, the *T_g_* temperature is the basis for plotting the viscosity curve. Transition temperature *T_g_* determined by the dilatometric method was used in the work, and then, using the relationship presented in Equation (5) [24], the viscosity curves of the studied glasses were plotted (Figure 4).
(5)$$log\;\eta =13.4-C_{1}\frac{T-T_{g}}{T-T_{g}+C_{2}}$$
where

*T_g_*—transition temperature at a viscosity of 10^13^ [dPa·s];*T*—temperature [°C];*C*_1_, *C*_2_—constant: *C*_1_—14.97; *C*_2_—278.

**Figure 4 materials-16-05789-f004:**
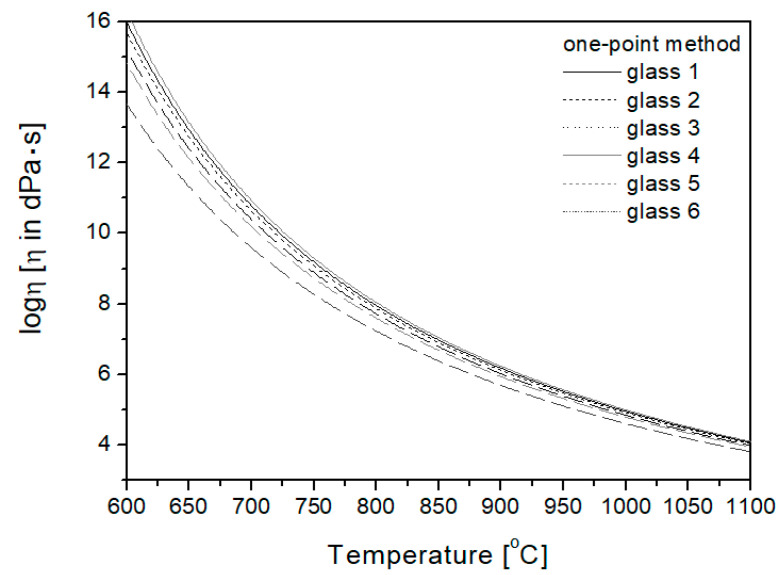
Viscosity curves plotted using the one-point method.

#### 3.4.2. Two-Point Method

In the two-point method, to determine the viscosity curve, in addition to the *T_g_* temperature, an additional point on the curve is taken into account in the range of higher temperatures at the viscosity of 10^4^ [dPa·s], i.e., the working point (*T_E_*) temperature. Taking this temperature into account allows more precise determination of the viscosity curve in the range in which the glass mass is characterized by greater fluidity [24].

In the viscosity calculations, temperature *T_E_* is taken into account to determine constant *C*_2_ (Equation (6)), which is then included in the calculations in Equation (5).

*C*_2_ = 0.588·(*T_E_* − *T_g_*)(6)

For the researched glasses, the *T_g_* temperature was used to plot the viscosity curves by the two-point method, determined dilatometrically and the hemisphere temperature, close to the working point temperature (*T_E_*), defined by hot stage microscopy as a fixed viscosity point, for which the viscosity value is 10^4.1^ [dPa·s]. The obtained curves are given in Figure 5.

Taking into account the additional point (working point temperature *T_E_*) on the viscosity curve is very important for glasses differing in the so-called technological length during forming. By comparing the viscosity curves of Glasses 1 and 6, determined by the one- and two-point method, it can be seen that there are changes in the inclination of the curves for the investigated glasses. For Glass 1, compared to Glass 6, there is a smaller difference between the transition temperature *T_g_* and the working point temperature (Figure 6). In the case of Glass 6, the curve determined by the two-point method has a greater slope and a smaller difference between *T_g_* and *T_E_ T* = 350 °C) than in the case of Glass 1 (Δ*T* = 450 °C). This means that Glass 6 is technologically “shorter” than Glass 1, which was not noticeable on the curves determined by the one-point method.

Comparing the viscosity curves in the range of from 10^3^ to 10^7^ [dPa·s] (Figure 7), plotted for Glasses 1 and 6 using the one- and two-point methods with fixed viscosity points determined experimentally (by hot stage microscopy—HSM), it can be seen that all the fixed viscosity points better fit into the surroundings of the curves drawn using the two-point method. This means that for aluminosilicate glasses, in order to actually present the course of viscosity changes in terms of greater liquidity of the glass mass (forming range), it is necessary to include temperature *T_E_* (*η* = 10^4^ [dPa·s]) in the calculations.

## 4. Conclusions

Based on the obtained research results and the conducted analysis, it was found that:Based on the analytical methods used to assess the viscosity of the glass mass in a wide temperature range, it was found that when selecting the appropriate calculation model, the basic chemical composition of the glass should always be taken into account, with particular emphasis on the share of amphoteric elements.It was justified that in order to plot the viscosity curve with the correct slope in the forming range, for aluminosilicate glasses it is appropriate to use the two-point method, based on fixed viscosity points for viscosities of 10^4^ [dPa·s] (working point) and 10^13^ [dPa·s] (transition temperature).Amphoteric metal ions (e.g., Al^3+^, Fe^3+^ and Mg^2+^) have a significant impact on the viscosity of the glass mass, and thus on the quality of the manufactured products. The way they are embedded in the glass structure can cause a modifying effect (coordination number 6) or a binding effect (coordination number 4). An increase in the content of metal ions in coordination number 4 results in the incorporation of these ions into the glass network, and thus its strengthening, which in turn causes an increase in the viscosity of the melt.In the range of high temperatures, with at viscosity in the range of from *η* = 10^3^ to 10^7^ [dPa·s], the modifying effect of the addition of basalt and cullet was found, manifested by a decline in the viscosity of the glass mass, accompanied by a drop in the *T_E_* temperature in the working range.In the process of producing aluminosilicate glass, a thorough analysis of the temperature parameters characterizing the raw material sets is of great importance.

## Figures and Tables

**Figure 1 materials-16-05789-f001:**
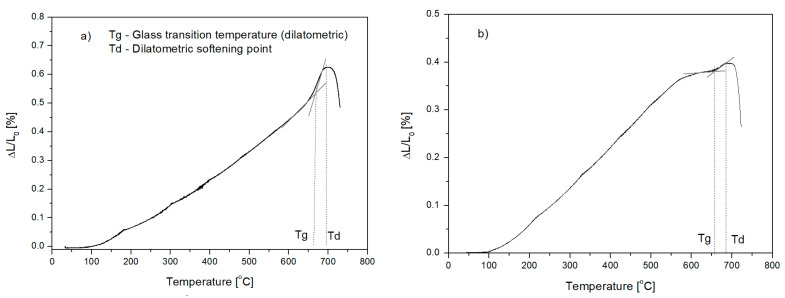
Dilatometric thermogram of glasses: (**a**) glass from Set 1, (**b**) glass from Set 4; *T_g_*—glass transition temperature, *T_d_*—dilatometric softening point.

**Figure 2 materials-16-05789-f002:**
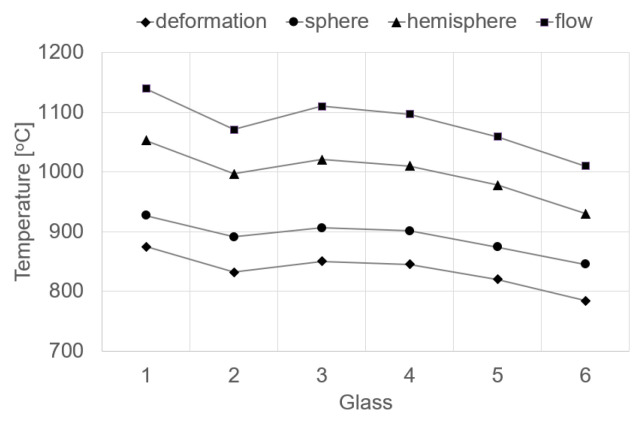
Fixed viscosity points of studied glasses determined by hot stage microscopy.

**Figure 3 materials-16-05789-f003:**
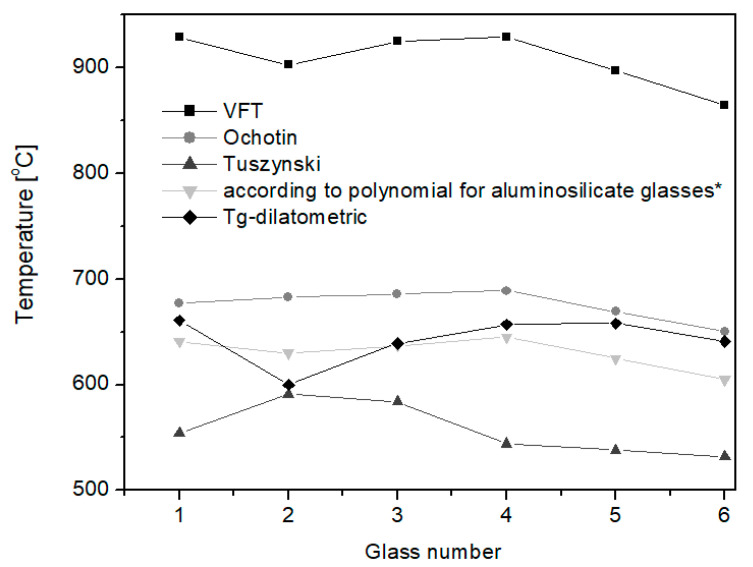
Comparison of *T_g_* temperature [°C] of *η* = 10^13^ [dPa·s] determined by various methods; * own elaboration [39].

**Figure 5 materials-16-05789-f005:**
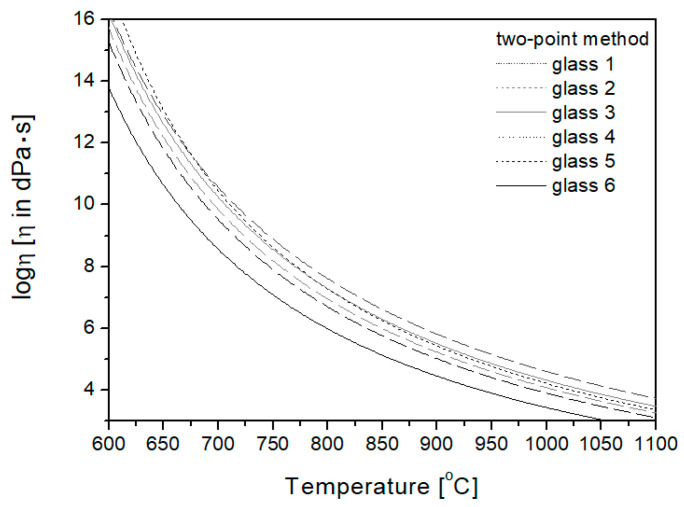
Viscosity curves plotted using the two-point method.

**Figure 6 materials-16-05789-f006:**
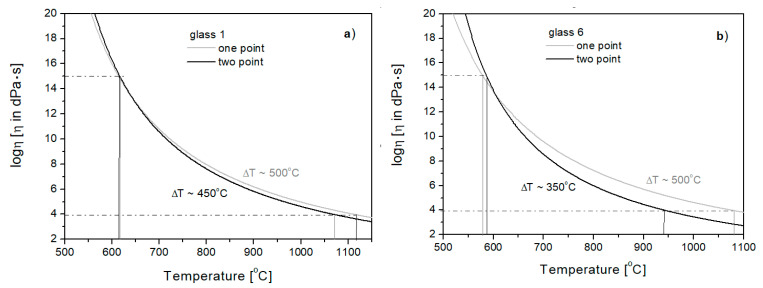
Viscosity curves of Glasses 1 (**a**) and 6 (**b**) determined by one- and two-point methods.

**Figure 7 materials-16-05789-f007:**
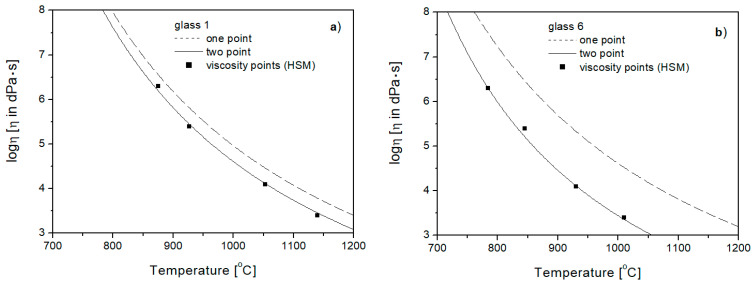
Viscosity curves of Glasses 1 (**a**) and 6 (**b**), in range of from 10^3^ to 10^7^ [dPa·s], determined by one- and two-point methods with fixed viscosity points.

**Table 1 materials-16-05789-t001:** Chemical composition and percentage of individual raw materials in sets, in weight %.

Oxide	Amphibolite	Basalt	Dolomite	Float Cullet
SiO_2_	55.58	41.01	3.27	71.89
Al_2_O_3_	15.23	14.22	1.37	0.59
CaO	6.27	11.01	59.34	9.28
Fe_2_O_3_	10.62	13.71	0.83	0.08
MgO	4.93	10.20	36.93	3.94
Na_2_O	5.58	5.43	0.37	13.72
TiO_2_	1.11	1.14	0.04	0.06
K_2_O	0.23	1.31	0.19	0.11
P_2_O_5_	0.10	1.02	0.06	<0.1
MnO	0.26	0.22	0.19	0.01
ZrO_2_	0.10	-	0.26	0.02
Raw Materials in Sets [wt%]				
Set	Amphibolite	Basalt	Dolomite	Float Cullet
1	100	-	-	-
2	70	-	10	20
3	85	-	5	10
4	40	50	-	10
5	30	50	-	20
6	20	50	-	30

**Table 2 materials-16-05789-t002:** Chemical composition of obtained glasses in wt%.

Oxides	Glasses					
	1	2	3	4	5	6
SiO_2_	56.28	53.86	54.73	46.97	48.59	50.23
Al_2_O_3_	14.53	11.20	13.25	13.21	11.84	10.47
CaO	6.27	11.92	9.01	9.31	9.30	9.30
Fe_2_O_3_	9.62	7.39	8.85	9.29	8.32	7.34
MgO	5.93	7.75	6.76	8.15	7.70	7.25
Na_2_O	5.58	6.72	6.01	6.35	7.79	9.24
TiO_2_	1.11	0.77	0.94	2.07	1.90	1.72
K_2_O	0.23	0.16	0.18	0.85	0.82	0.79
P_2_O_5_	0.10	0.00	0.00	0.43	0.43	0.43
MnO	0.26	0.23	0.27	0.15	0.13	0.11
ZrO_2_	0.10	0.00	0.00	3.21	3.17	3.13

**Table 3 materials-16-05789-t003:** Ranges of content of individual oxides in investigated glasses in wt%.

Oxides	SiO_2_	Al_2_O_3_	CaO	MgO	Fe_2_O_3_	Na_2_O	K_2_O
wt%	46–56	10–15	6–12	5–8	7–11	5–9	0–1

**Table 4 materials-16-05789-t004:** Transition (*T_g_*) and dilatometric softening (*T_d_*) temperatures of glasses in °C.

Glass	1	2	3	4
*T_g_*	661	600	639	657
*T_d_*	695	636	642	685

**Table 5 materials-16-05789-t005:** Fixed viscosity points determined by hot stage microscopy (*η* in [dPa·s]).

Glass	Characteristic Temperatures [°C]			
	Deformation *logη* = 6.3	Sphere *logη* = 5.4	Hemisphere *logη* = 4.1	Flow *logη* = 3.4
1	875	927	1053	1140
2	832	891	997	1071
3	850	906	1021	1110
4	845	901	1010	1097
5	820	874	978	1059
6	784	845	930	1010

**Table 6 materials-16-05789-t006:** Calculated temperature [°C] at specified viscosities of 10^13^ and 10^4^ [dPa·s] according to Vogel–Fulcher–Tammann.

Viscosity [dPa·s]	Glass					
	1	2	3	4	5	6
	Temperature [°C]					
10^13^	929	903	925	929	897	865
10^4^	3149	3281	3269	3412	3259	3118

**Table 9 materials-16-05789-t009:** Coefficients for determining temperature [°C] at specified viscosity from 10^3^ to 10^13^ [dPa·s] according to Tuszynski method [13].

Viscosity [dPa·s]	A_1_	A_2_	A_3_	A_4_	A_5_
10^3^	16.73	23.2	0.63	9.6	−6.14
10^4^	13.63	19.5	3.68	9.7	−3.86
10^5^	11.79	16.8	5.54	10.5	−3.58
10^6.5^	9.70	14.3	7.51	11.0	−2.49
10^7^	9.03	13.4	7.85	10.5	−1.33
10^8^	8.12	12.4	8.59	10.0	−0.59
10^9^	6.57	10.2	8.87	8.9	4.52
10^10^	7.18	10.7	9.82	8.8	−1.43
10^11^	6.75	10.1	10.90	8.1	−1.24
10^12^	6.46	10.0	9.66	6.7	−0.97
10^13^	6.10	9.9	9.25	5.3	−0.04

**Table 10 materials-16-05789-t010:** Calculated temperature [°C] at specific viscosities of 10^13^ and 10^4^ [dPa·s] according to Tuszynski method.

Viscosity [dPa·s]	Glass					
	1	2	3	4	5	6
	Temperature [°C]					
10^13^	554	591	584	544	538	532
10^4^	1067	1046	1080	983	969	954

**Table 11 materials-16-05789-t011:** Main content ranges of oxides, wt% [19].

Oxide	SiO_2_	Al_2_O_3_	CaO	MgO	Fe_2_O_3_	R_2_O *
wt%	45–60	8–20	10–25	3–15	2–10	4–6

* R-alkali oxide (Na, K).

**Table 12 materials-16-05789-t012:** Calculated temperature [°C] at specified viscosities of *η* = 10^13^ and *η* = 10^4.0^ [dPa·s], according to polynomial for aluminosilicate glasses.

Viscosity [dPa·s]	Glass					
	1	2	3	4	5	6
	Temperature [°C]					
10^13^	641	630	637	645	625	605
10^4^	1070	1012	1038	1026	994	943

## Data Availability

Resources available on request.

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
