# Peer review of "Experimental vs. Theoretical Viscosity Determination of Aluminosilicate Glasses"

_materials, 2023, doi:10.3390/ma16175789_

Round 1

Reviewer 1 Report

The authors applied various theoretical models to investigate the relationship between viscosity and temperature using experimental data of six aluminosilicate glasses. The overall information provided satisfies the scope of the Journal. However, following comments should be addressed before consideration of acceptance.

1. The authors claimed that their model together with the Tuszynski method can provide more accurate prediction of the viscosities comparing with rest models. It thus questionable that how reliable the new-establish model will be. Is it over fitted by the glass compositions in this work? The authors should clarify this and add to the discussion part.

2. In addition, the authors stated “….the appropriate calculation model should always be selected for the type of glass for which the calculations are carried out…” This rises another question, is the proposed equation possible to be used in other aluminosilicate glass systems? The authors should clarify this and add to the discussion part.

3.  In Figure 7, the legend says one point is presented in dashed line, however, none of the two lines are dashed. This should be corrected.

4. Various typos have been found, following are some of them. The authors should carefully perform a proof reading.

a) Tg, TE should be with subscripts.

b) Definition of TE in page 12 should be pushed backward after temperature.

c) In page 13, “… in the range of 103¸107” should be 103~107. This has been observed in various place in the manuscript.

Reviewer 2 Report

Interesting manuscript with minor revision:

1-Title should be rewritten,

2-Results and Analysis need comparison with other data:

A-Physica Status Solidi (a) 177(2), 439-444 (2000)

3-Conclusion should be quantitative.

Minor Editing of English language required

Reviewer 3 Report

The article studies possible ways of glass viscosity calculations using different methods. It is very important issue of glass technology because experimental analysis of viscosity changes is difficult. However, the article has a number of weak and arguable points:

1. Article should be formatted according to Journal’s requirements (for example, lines 33-34 spacing)

2. The text is hard to read. Sentences should be shorter.

3. Viscosity units should be Pa·s or Pa*s but not Pas. Viscosity was measured in poises once, but now it should be in SI units. So viscosity values should be recalculated from dPas to Pa*s.

4. Statement that MgO is amphoteric is really doubtful because properties of Ca and Mg are quite close. Authors should strongly prove that MgO is amphoteric. Otherwise the whole article is based on the wrong statement.

5. The same is for coordination numbers 4 and 6 for Mg – their existence should be proved.

6. Authors should add chemical (oxide) compositions of the initial raw materials: amphibolite, basalt, glass, and dolomite.

7. Okhotin’s name is spelled wrong. For example: Okhotin M.V., Andryukhina T.D. Calculation of the viscosity of low-alkali and nonalkali glasses // Glass and Ceramics. 1970. Т. 27. № 1. С. 16-17.; Okhotin M.V., Raevskaya E.I., Tuzikov A.I. An experiment on the industrial determination of the viscosity of the glass melt in the feeder of a tank furnace // Glass and Ceramics. 1968. Т. 24. № 12. С. 682-684.

8. Many of described methods cannot be applied to any type of glass. For example, M.V. Okhotin developed a number of methods. Speking about the article, there are two appropriate methods:

- For multicomponent alkaline glasses, which is limited by following, wt.%: Na20 – 12-16, СаО – 5-12, MgO – 0-5, Аl2O3 0-5.

- For low-alkaline glasses, which is limited by following, wt.%: Na20 – 0-5; СаО - 10,4-15,4; MgO - 4,1; Аl2О3 - 13,5-18,5, and F - 4 over 100%.

Other compositions will give inappropriate errors to the resulting viscosity. Authors should carefully check whether methods they choose can be used for their glasses.

9. Authors should consider a new Mauro–Yue–Ellison–Gupta–Allan (MYEGA) viscosity expression and how it corresponds to their hypothesis.

10. What is the purpose of Table 3? Why do authors add it?

11. Tg is usually called a glass transition temperature, but not transformation.

12. Why is coordination number called LK 6? What is LK?

13. Composition of Set 3 (80 wt% amphibolite, 5 wt% dolomite, 10 wt% cullet) is wrong.

14. Name of Table 6 is repeated twice.

15. Equation 2 is wrong: x, y, and z shouldn’t be subscriptive.

16. Conclusion for section 3.3.3 (Tuszynski method) should be added.

17. Name of Table 12 should be written properly.

18. Name of grey-triangle curve (Figure 3) should be revised.

19. The authors should add references describing the full calculations for all methods used in the article.

20. The authors should add the results of hot stage microscopy which they are often referring.

21. There are no dotted curves in Figure 7. It should be revised.

Thus, the article has a practical and theoretical importance, but it should be strongly reworked to be published in the Journal.

The text is hard to read. Sentences should be shorter.

Round 2

Reviewer 3 Report

The article became much better. However, I’d suggest few corrections before it’ll be published:

- The word “Theoretical” in the article’s title shouldn’t begin with capital letter.

- Remark 4. According to chemical definition, amphoteric oxides are compounds that react with both acids and bases to form salts and water. Hence, the term “amphoteric” isn’t suitable in our case. MgO cannot interact with bases. The property that authors try to describe with “amphoteric” (that MgO could act both as network modifier and network builder) isn’t described with this term. So, it should be revised.

- Remark 5. References on the ability of MgO to create both four-fold and six-fold cells should be added in the text. The same is for Remark 4 ( references proving the modifying and binding effect).

- Remark 7. As a native speaker I can assure authors that the correct spelling is “M.V. Okhotin”, and I strongly recommend to use it.

- Remark 8. Authors use calculation methods which are inappropriate for their glasse. For example, Okhotin’s method that they use is only for multicomponent alkaline glasses, which is limited by following, wt.%: Na20 – 12-16, СаО – 5-12, MgO – 0-5, Аl2O3 0-5. So, authors should add requirements for every method and state when they expect errors because of unsuitable chemical composition.

- Statement from authors’ answer No. 10 should be added into the text.

When these points will be reworked the article can be published in the Journal.

The article became much better. However, I’d suggest few corrections before it’ll be published:

- The word “Theoretical” in the article’s title shouldn’t begin with capital letter.

- Remark 4. According to chemical definition, amphoteric oxides are compounds that react with both acids and bases to form salts and water. Hence, the term “amphoteric” isn’t suitable in our case. MgO cannot interact with bases. The property that authors try to describe with “amphoteric” (that MgO could act both as network modifier and network builder) isn’t described with this term. So, it should be revised.

- Remark 5. References on the ability of MgO to create both four-fold and six-fold cells should be added in the text. The same is for Remark 4 ( references proving the modifying and binding effect).

- Remark 7. As a native speaker I can assure authors that the correct spelling is “M.V. Okhotin”, and I strongly recommend to use it.

- Remark 8. Authors use calculation methods which are inappropriate for their glasse. For example, Okhotin’s method that they use is only for multicomponent alkaline glasses, which is limited by following, wt.%: Na20 – 12-16, СаО – 5-12, MgO – 0-5, Аl2O3 0-5. So, authors should add requirements for every method and state when they expect errors because of unsuitable chemical composition.

- Statement from authors’ answer No. 10 should be added into the text.

When these points will be reworked the article can be published in the Journal.
